# Different Ecological Niches of Poisonous *Aristolochia clematitis* in Central and Marginal Distribution Ranges—Another Contribution to a Better Understanding of Balkan Endemic Nephropathy

**DOI:** 10.3390/plants12173022

**Published:** 2023-08-22

**Authors:** Ivan Brzić, Magdalena Brener, Andraž Čarni, Renata Ćušterevska, Borna Čulig, Tetiana Dziuba, Valentin Golub, Irina Irimia, Bojan Jelaković, Ali Kavgacı, Mirjana Krstivojević Ćuk, Daniel Krstonošić, Vladimir Stupar, Zlatko Trobonjača, Željko Škvorc

**Affiliations:** 1Municipality of Bebrina, 35254 Bebrina, Croatia; 2School of Medicine, University of Rijeka, Braće Branchetta 20/1, 51000 Rijeka, Croatia; zlatko.trobonjaca@uniri.hr; 3Faculty of Forestry and Wood Technology, University of Zagreb, Svetošimunska 23, 10000 Zagreb, Croatia; mbrener@sumfak.hr (M.B.); dkrstonosic@sumfak.hr (D.K.); zskvorc@sumfak.hr (Ž.Š.); 4Institute of Biology, Research Center of the Slovenian Academy of Sciences and Arts, Novi Trg 2, SI 1000 Ljubljana, Slovenia; carni@zrc-sazu.si; 5School for Viticulture and Enology, University of Nova Gorica, Vipavska 13, SI 5000 Nova Gorica, Slovenia; 6Institute of Biology, Faculty of Natural Sciences and Mathematics, Ss. Cyril and Methodius University, 1000 Skopje, North Macedonia; renatapmf@yahoo.com; 7Faculty of Medicine, Josip Juraj Strossmayer University of Osijek, Josipa Huttlera 4, 31000 Osijek, Croatia; bornaculig@gmail.com; 8M.G. Kholodny Institute of Botany, National Academy of Sciences of Ukraine, 01004 Kyiv, Ukraine; tdziuba2014@gmail.com; 9Institute of Ecology of the Volga Basin, Samara Scientific Center of the Russian Academy of Sciences, 445003 Togliatti, Russia; vbgolub2000@mail.ru; 10Faculty of Biology, Alexandru Ioan Cuza University of Iași, 20A Carol I Blvd., 700506 Iași, Romania; iblaj2002@yahoo.com; 11Department of Nephrology, Hypertension, Dialysis and Transplantation, University Hospital Center Zagreb, School of Medicine University of Zagreb, 10000 Zagreb, Croatia; jelakovicbojan@gmail.com; 12Burdur Food Agriculture and Livestock Vocational School, Burdur Mehmet Akif Ersoy University, 15030 Burdur, Turkey; akavgaci@mehmetakif.edu.tr; 13Department of Biology and Ecology, Faculty of Science, University of Novi Sad, Trg Dositeja Obradovića 2, 21000 Novi Sad, Serbia; mirjana.krstivojevic@dbe.uns.ac.rs; 14Department of Forest Ecology, Faculty of Forestry, University of Banja Luka, S. Stepanovića 75A, 78000 Banja Luka, Bosnia and Herzegovina; vladimir.stupar@sf.unibl.org

**Keywords:** poisonous plant, niche width, ethnoecology, environmental conditions, vegetation plots, ecological specialization index, traditional agroecology praxis

## Abstract

*Aristolochia clematitis* L. is a perennial herbaceous plant distributed throughout Europe, Asia Minor and Caucasus. It has been used as a medicinal plant since antiquity but not in recent times because it contains poisonous aristolochic acid, causing progressive kidney failure. The aim of this work was to study *Aristolochia clematitis* ecology on the basis of vegetation plots from the European Vegetation Archive, and to investigate the differentiation of its ecological niche using a co-occurrence-based measure of ecological specialization (ESI). The ecological niche was studied on three spatial scales: on the entire distribution area, its differentiation across 200 × 200 km grid cells and the differences between three central and three marginal regions. Our results suggest that *Aristolochia clematitis* has a very broad ecological niche occurring in a range of different habitats and climatic conditions, with a trend of a niche width decrease with the distance from the geographical center. The plant prefers more stable communities with less anthropogenic influence moving towards the margin of the distribution area. Specialization towards the marginal area is a result of evolutionary history, which refers to the recent anthropogenically induced spread from its original home range. A high incidence of *Aristolochia clematitis* in the vegetation of arable lands and market gardens as well as anthropogenic herbaceous vegetation in the distribution center corresponds to the geographical incidence of Balkan Endemic Nephropathy.

## 1. Introduction

The concept of a niche in ecology refers to the specific role and position that a species holds within its ecosystem and can be defined as the position and breadth of a species’ distribution along various niche axes [1]. Niche theory assumes that a species has a fundamental niche [2] or physiological niche [3] in relation to some environmental gradient and in the absence of competitors. In the presence of competitors, the species niche is reduced to a realized or ecological niche. The impacts of competitors reduce the width of the realized niche and may shift its position [2,3]. 

The ecological response of organisms to different environmental factors varies across a species’ geographical range. This variation may be the result of local adaptation to environmental conditions and may influence both fundamental and realized niche differentiation across the species’ geographical range. The differentiation of the realized niche within the same global fundamental niche is the result of processes, such as variation in biotic interactions in different parts of the species’ geographical range (e.g., facilitation, competitive displacement), habitat compensation processes (e.g., compensation of ecophysiological stress where species compensates the deterioration of physiological living conditions at the edge of their range by a change in the niche position) or species’ dispersal limitation [4,5,6,7,8]. 

Usually, two parameters are used to describe the differentiation of the ecological niche within the geographical range of the species niche width (range of environmental conditions where it occurs) and niche position (ecological optimum or species’ maximum probability of presence within its realized niche). There is a wide range of metrics for the estimation of species’ niche width [9], all having their advantages and disadvantages. In the classic approach, the niche width is determined as the range of values along each of the particular environmental gradients that a species is able to utilize [10,11]. A species can have very narrow ecological niches on one particular measured gradient (specialists) but wider niches on some other (generalist) [12]. There is also an alternative approach, used in this study, which quantifies the species niche width using indirect estimations of the diversity of habitats in which the species occurs. Fridley et al. [13] introduced a method based on a principle that the niche width of a species corresponds to the pattern of its co-occurrence with other species. Habitat specialists occur in different localities with the same group of species, indicating that different localities are likely to be ecologically similar. Habitat generalists co-occur with a wide range of species across localities, indicating that these localities are ecologically heterogeneous. In this way, it is possible to measure species specialization without data about environmental factors at sites where the species are recorded, but it takes into consideration many environmental factors. Based on the original method of Fridley et al. [13], some methodological modifications were proposed [14,15,16], and many studies have applied this approach for a broad range of research questions [6,11,17,18]. Finally, inspired by the original method of Fridley et al. [13], Zelený and Chytrý [19] introduced the Ecological Specialization Index (ESI) as a measure of differences in species composition among samples containing the focal species using multiplicative beta diversity. 

*Aristolochia clematitis* L. (*Aristolochiaceae*) is a perennial herbaceous plant that can grow up to 1 m. It often propagates vegetatively, through fragile rhizomes. The original area of distribution probably includes Southern Europe, Asia Minor and the Caucasus, and it secondarily spread to Central and Eastern Europe. It had probably already begun to spread significantly in the Middle Ages, both as a weed (especially in vineyards) and as a result of cultivation for medical purposes [20,21]. *A. clematitis* usually grows in warm, sunlit places, with nutrient-rich soils, in light floodplain forests, on the banks of watercourses, on embankments, wastelands, scrubby slopes, in vineyards and beside road and railway embankments [20,21]. It is considered as a diagnostic species for riparian gallery forests (*Alno glutinosae-Populetea albae*) and tall-herb semi-natural perennial vegetation on disturbed forest edges, nutrient-rich riparian fringes and in forest clearings (*Epilobietea angustifolii*) [22].

*Aristolochia clematitis* has been used as a medicinal plant since antiquity, but today, it is forbidden for medical use due to its content of aristolochic acid, formally classified as a human carcinogen [23]. Balkan Endemic Nephropathy (BEN) is a chronic kidney disease associated with urothelial carcinomas that affects residents of rural farming villages located along tributaries of the Danube River in Bosnia and Herzegovina, Bulgaria, Croatia, Romania and Serbia [24,25,26]. Fifty years ago, Ivić proposed a role for chronic *Aristolochia* poisoning in the etiology of BEN [27]. He observed that seeds from these plants, which grew abundantly in local wheat fields, were mixed with wheat grain during the harvesting process. In a study, which included farmers who lived in 54 different villages in Bosnia, Croatia and Serbia, 86.4% of subjects with BEN reported that they had observed *Aristolochia clematitis* in wheat fields 20–30 years ago [28]. Research projects conducted in Croatia definitively confirmed that aristolochic acid is the most important risk factor and causative agent for BEN [26,29,30].

An important feature of BEN is focal, mosaic distribution. It has been reported in restricted areas in Balkan countries, and, even in these focal areas, not all villages were affected. It is questioned whether differences in the incidence of BEN could be explained with the species geographical range and with the ecological niche of *Aristolochia clematitis*.

Understanding the patterns of ecological niche differentiation and their underlying mechanisms is essential for understanding species’ abundance and distribution patterns from the past, as well as for predicting their responses to environmental changes [31,32]. The differentiation of the realized niche is dominantly focused on restricted geographical extents, comparing central and peripheral populations [31,33], while there are not many studies of its differentiation along the geographical range of the species [5,6]. *Aristolochia clematitis* is a suitable target species for the study of ecological niche differentiation because it occurs over a large geographical area in very different habitats. Taking into account the insufficient knowledge of the species ecology, its past and present distribution, as well as its great importance as a cause of BEN, the aim of this work is to study the ecology of that species in more detail on the basis of vegetation plots from the European Vegetation Archive, and to investigate the width of the ecological niche and its differentiation on the entire area of the species distribution, using a co-occurrence-based measure of ecological specialization (ESI).

## 2. Materials and Methods

In order to investigate the ecological niche of *A. clematitis*, georeferenced vegetation plots from the European Vegetation Archive were used. Ecological conditions are described for each plot using several parameters: bioclimatic variables, habitat type, vegetation type, ecological indicator values and disturbance indicator values. The analysis was performed in three steps: (i) description of the ecological niche using the above indicators in the entire studied area, (ii) exploration of the ecological niche differentiation using the Ecological Specialization Index (ESI), and (iii) description of the differences in the ecological niche between the margin and center of the distribution area.

### 2.1. Vegetation Plots

The initial dataset consisted of 3239 vegetation plots (relevés) with the occurrence of *A. clematitis* from the European Vegetation Archive (EVA; [34], data exported on 9 November 2022). We removed records of non-vascular plants and merged the same species in different vegetation layers to ensure that each species occurred only once in each plot.

The database was geographically stratified to reduce oversampling of similar habitats within some regions [35]. Stratification was performed by assigning the plots to grid cells of approximately 200 km × 200 km that were subsequently used for the analysis. Grid cells were based on Universal Transverse Mercator (UTM) grid system derivative, i.e., Military Grid Reference System (MGRS) with a square side length of 100 km. We performed merging of four adjacent 100 km squares to obtain 200 km × 200 km squares. Squares that were, prior to the merging, adjacent to a UTM Grid Zone Junction and, therefore, clipped were merged with the next full 100 km square with the same Grid Zone Designator. The same was carried out in the case when MGRS 100 km square crossed a latitude band boundary. Such spatial resolution, although coarse, accounts for poorly sampled areas in our dataset since a small number of plots cannot properly represent the habitat diversity of a particular area.

Within each MGRS grid square, we calculated Bray–Curtis dissimilarity among all pairs of plots (using log-transformed percentage covers in species composition data) and applied heterogeneity-constrained random (HCR) resampling procedure to resample the optimal subset of plots retaining maximum mean pairwise dissimilarity among selected plots [36]. Since each MGRS square may contain plots belonging to a wide range of habitats, we accepted recommendation of Wiser and De Cáceres [37], that more plots are selected from squares that have higher compositional heterogeneity. The minimum and maximum number of selected plots was 5 and 20, respectively. The resulting dataset, called the ‘whole dataset’ throughout this study, contains 851 vegetation plots and 2168 species (Figure 1, Appendix A). This dataset is used for the description of the whole ecological niche, without considering niche differentiation. 

### 2.2. Ecological Specialization Index and Niche Differentiation

Ecological Specialization Index (ESI) of *A. clematitis* was calculated for each of the MGRS grid cells following procedure of Zelený and Chytrý [19]. All calculations were performed in the R program (R Core Team 2017) using ‘theta’ library (https://github.com/zdealveindy/theta, accessed on 15 February 2023). As this procedure implies at least 10 occurrences of a given species per stratum. ESI was calculated only for 45 MGRS grid cells with at least 10 plots, which remained after HCR resampling (Figure 1). The final dataset for the analysis of niche differentiation contains 718 vegetation plots and 1918 species.

To study the difference between the central and marginal parts of species range in more detail, six regions were chosen (three in marginal and three in central area). In each region, three MGRS grid cells were merged in longitudinal direction (Figure 1).

### 2.3. Data for Niche Description

All plots were classified based on their species composition and species cover into EUNIS habitat types (European Nature Information System, 3rd hierarchical level) using the EUNIS-ESy expert system v. 2024-06-01 [38]. To avoid a large number of outliers as well as to obtain a clearer pattern, plots were merged into a smaller number of broader units (1st and 2nd level). Furthermore, plots were classified into vegetation classes using EuroVegChecklist expert system [22].

To describe macroclimatic gradients, each plot was characterized based on its geographical coordinates using 19 bioclimatic variables obtained from Chelsa v2.1 [39]. To quantify environmental factors on the micro scale, we used Ellenberg-type indicator values for light, temperature, moisture, reaction and nutrients [40]. Disturbance indicator values were used to estimate anthropogenic impact on studied vegetation types [41]. Four main continuous indicators were used: disturbance severity (mean magnitude of disturbance events—proportion of aboveground biomass killed by disturbance), soil disturbance (proportional increase in cover of bare ground by furrowing or soil turning), mowing frequency (mean frequency of cutting of plant biomass) and grazing pressure (severity of grazing—proportion of aboveground biomass killed by grazing). Unweighted mean Ellenberg-type and disturbance indicator values were calculated for each plot using JUICE v7.1 software [42].

Geographical center of all analyzed vegetation plots was calculated using ‘Mean coordinate(s)’ algorithm, while distance of the centroid of each MGRS grid square from the geographical center was calculated using ‘Distance matrix’ algorithm, both in QGIS v3.28 software.

### 2.4. Statistical Analysis

Descriptive statistical parameters and box-and-whiskers diagrams were used to describe the ecological niche and differences between regions. The significance of differences in ESI values as well in number of habitats and vegetation classes between central and marginal MGRS200 cells was tested using nonparametric Mann–Whitney U test (*p* < 0.01). The significance of differences in Ellenberg-type and disturbance indicator values between central and marginal MGRS200 cells was tested using nonparametric Kruskal–Wallis H test and multiple comparisons of mean ranks (*p* < 0.05). 

Relationships between ESI and distances of MGRS200 centroids from the geographic center were explored using Pearson’s r correlation coefficient (*p* < 0.001). Median values of all environmental variables in each MGRS200 square as well as percentage of EUNIS habitat types and vegetation classes were calculated. ESI was related to these medians as well as to niche width for each environmental gradient using Spearman correlations (*p* < 0.01). Niche width for each environmental gradient was estimated using quartile range of these variables in MGRS200 squares. All these calculations were performed in Statistica v14.0 software (TIBCO Software Inc., Palo Alto, CA, USA, 2020).

To explore the main patterns in the species composition of the vegetation plots in central and marginal regions as well as their relationship with environmental gradients, a detrended correspondence analysis (DCA) was used. DCA with passive projection of environmental variables and EUNIS habitat types was performed using the R package ‘vegan’ (https://cran.r-project.org/web/packages/vegan, accessed on 15 February 2023), operated through the JUICE v7.1 software [42].

## 3. Results

### 3.1. Characteristics of Ecological Niche in the Whole Studied Area

*A. clematitis* occurs in a wide range of climatic conditions, with a mean annual air temperature range between 0.8 and 18.1 °C and annual precipitation range between 276 and 2234 kg m^−2^ (Appendix A). It prefers moderately warm habitats and appears in a wide range of soil moistures and nutrients (Appendix A). Such a wide ecological niche is confirmed by the large number of habitats (53 EUNIS habitat types, level 3) as well as the large number of vegetation classes in which the species occurs (26 classes). Generally, it most often appears in riparian forests and on the edges and clearings next to these forests. It also very often occurs as a part of anthropogenic herbaceous vegetation. Although it is more often associated with moist or wet habitats, it can also be found in dry habitats, such as semi-dry and dry grasslands (Appendix A).

### 3.2. Differentiation of the Ecological Niche

A significant difference in ESI among the MGRS200 cells was determined, ranging from 1.88 to 6.59. The ESI pattern is quite heterogeneous across the distribution area, but a significant relationship is generally observed between the distance of the MGRS200 centroid from the geographic center and the ESI (Figure 2). In other words, there are considerable differences even between neighboring MGRS200 cells, but generally, *A. clematitis* is more specialized in squares that are further from the geographic center (Figure 3).

MGRS200 cells where *A. clematitis* is more specialized (higher ESI) have lower ecological optimums for light and soil reaction. The relationships between ESI and the average disturbance regime on individual MGRS200 cells are even more significant; a higher ESI (narrower ecological niche) is related to a lower disturbance frequency and lower soil disturbance (Figure 3 and Appendix A). In MGRS200 cells where *A. clematitis* is more specialized (higher ESI), it occurs more often in forest habitats and less often in man-made habitats (Table 1, Figure 3 and Appendix A).

It is evident that ESI is significantly related to the width of the gradient for habitat moisture and all types of disturbance, indicating that the reduction in ESI is associated with a wider ecological niche for moisture and disturbance (Table 1).

### 3.3. Differences between Margin and Centre of the Distribution Area

In the studied marginal regions, *A. clematitis* is significantly more specialized (*p* < 0.001). The MGRS200 cells belonging to marginal regions had ESI values from 3.4 to 4.9 (mean 4.3), and those belonging to the central regions from 1.8 to 3.8 (mean 2.9). This is also clearly visible from the number of habitats and vegetation classes. In central regions, *A. clematitis* occurs in a greater number of EUNIS level 3 habitats (*p* = 0.045) and vegetation classes (*p* = 0.029) compared to marginal ones. 

*A. clematitis* in peripheral regions occurs more often in forest habitats (*p* = 0.003, Figure 4 and Figure 5), so the ecological optimum on the light gradient is shifted towards lower values (Figure 6). In the central regions, the overall ecological niche is wider, primarily because *A. clematitis* occurs more often in anthropogenic habitats there (Figure 4 and Figure 5), so the ecological optimum is shifted towards a higher disturbance frequency and stronger soil disturbance (Figure 7). This mostly refers to the greater representation of arable lands and market gardens (such as communities of Papaveretea rhoeadis and Digitario sanguinalis-Eragrostietea minoris) as well as annual anthropogenic herbaceous vegetation (such as communities of Sisymbrietea).

## 4. Discussion

Our results suggest that *A. clematitis* generally has a very broad climatic niche, avoiding only the most xerothermal Mediterranean and northern hemiboreal and boreal climates (Appendix A). This is somewhat unusual considering that the species originally spread in warmer areas south of the Alps and is an indication that it has successfully adapted to very different climates. In this climatically very different area, it occurs in a range of different habitats, from swamps and grasslands, riparian and mesic forests to arable lands, which is in accordance with the existing literature [20,21]. Regardless of its generally wide niche, there is a visible difference in the width of the ecological niche on different environmental gradients. Thus, for example, it occurs in a much wider range of soil moisture and nutrient values compared to the other studied gradients (Figure 6). It is very common that the niche width varies along different gradients, so on some, the species behaves as a specialist and on others as a generalist.

Our results also suggest that the realized niche space occupied by *A. clematitis* varies across distribution ranges, with a clear trend for niche width to decrease with distance from the geographic center (Figure 2 and Figure 3). It has often been hypothesized that species on their distribution margin have narrower ecological niches than in the center [43], but the pattern is much more complex in different species, and there is probably no general rule [5,6], mostly because, in many species, geographic, environmental and historical centrality and marginality do not overlap [44,45]. Specialization towards the edge of the distribution area in *A. clematitis* is a result of its evolutionary history, which refers to recent anthropogenically induced spread from its home range. In the new regions, it is still not adapted to dramatically different climatic conditions. Šilc et al. [17] presented very similar findings comparing weed species’ ecological niche on north–south gradients. The character and degree of differences between populations close to the species’ range limits and those in the central part of their distribution are some of the fundamental questions in ecology [45]. The ‘Centre-periphery’ hypothesis is a common hypothesis explaining patterns of these differences. It states that genetic variability, individual fitness and population demography of a species decrease from the center to the edge of its geographic range [43], and this could lead to “ecological marginality” [45] and a reduced ecological amplitude or niche width [46].

Through this descriptive study, we cannot fully discriminate the reasons underlying the patterns of niche variation between central and peripheral populations because variation may be associated with a range of factors. According to Zelený and Chytrý [19], calculated ESI values are estimates of the realized niche width resulting from the interaction of three factors: (i) species fundamental niche, (ii) availability of suitable habitats for the focal species in the study area, and (iii) biotic interactions with other species.

### 4.1. Fundamental Niche

The species’ fundamental niche is defined by its physiological constitution created during evolution and can be changed through local adaptation to new environmental conditions. However, in our case, the marginal populations of *A. clematitis* spread to a new area relatively recently, and, probably, no significant local adaptation occurred.

### 4.2. Availability of Suitable Habitats

Our analyses show that there is an ecological shift in marginal populations that occurs to a much lesser extent in dry, open anthropogenic habitats (Figure 6 and Figure 7). In this way, they have both a narrower ecological niche on the gradients of light and moisture and a shift in the ecological optimum towards wetter and less open habitats (Table 1, Figure 6). They also have a narrower ecological niche for disturbance, as well as a shift in the ecological optimum towards a lower disturbance frequency and a lower soil disturbance (Table 1, Figure 7). Šilc et al. [17] also found that weed species that evolutionarily adapted to disturbed and warm sites in the south have narrower ecological niches in northern areas, although the level of agricultural disturbance along the gradient is similar. However, in their case, the species towards the range margin maintained a weed strategy and specialized in locally warmer and drier habitats, as a response to changed macroclimatic conditions. In our case, *A. clematitis* specialized to habitats with less anthropogenic pressure. Other environmental factors seem to have a smaller impact, although this specialization in itself led to an ecological shift towards wetter and darker habitats. 

We consider that the availability of anthropogenic habitats in marginal areas did not significantly affect the ecological niche shift. Specifically, all types of anthropogenic habitats where *A. clematitis* occurs are available, both in the marginal and central parts of the distribution area [38], as well as most of the corresponding vegetation alliances [47]. However, although it probably spread to new areas anthropogenically [20,21], for some reason, it prefers more stable communities with less anthropogenic influence in these climatically different areas. In fact, this pattern is in accordance with the general rule that specialists occur in more stable environments [48], so towards range margins, it stays in more stable communities such as forests (Table 1, Appendix A). Such regional differences in the ecological niche may result from time-lagged range expansion from the Middle Ages and disequilibrium with the current climate in the new area [7]. 

It seems that, in the past, under favorable conditions, in the center of present distribution, the species became more competitive and expanded its ecological niche to various anthropogenic habitats. It is possible that these conditions during the Neolithic, when the climate became warmer and drier, combined with the spread of agriculture, stimulated many of today’s weed species that were originally specialists in other habitats (e.g., forests, shrublands) to spread into anthropogenic habitats and become generalists [17]. Suitable climatic conditions combined with the rapid expansion and spread of human-modified habitats provided opportunities for plant species to expand their ecological niche [49]. Similarly, current global climate change could stimulate *A. clematitis* in marginal areas to expand its ecological niche towards different anthropogenic habitats.

### 4.3. Biotic Interactions

Biotic interactions, such as competition or facilitation, are important determinants of a species’ realized niche, although they are difficult to measure directly. Biotic interactions depend on the composition of the regional species pool, namely the presence or absence of competitors that would narrow the realized niche or facilitators that would broaden it [50]. In the north, there is a smaller weed species pool but a larger number of weed species per plot [17,51], so it is possible that strong competition of weed species excludes *A. clematitis* from such habitats, reducing its ecological niche at the marginal part of the distribution area. 

Furthermore, in areas with high regional (gamma) diversity (e.g., Southern France, Balkans), *A. clematitis* has a wider ecological niche compared to areas in the north (Figure 2) where regional diversity is lower. It is not in accordance with the theory, which claims that the restriction of a species’ realized-niche width through competition increases with regional diversity [52]. However, this is consistent with the findings of Manthey et al. [53] who showed that there was no such effect of the size of the species pool (species competition) on the restriction of the species’ realized niche width and that plant species are more constrained by environmental conditions than by competition.

### 4.4. Methodology Strengths and Weaknesses

The method we used to quantify the ecological niche of *A. clematitis* critically depends on the quality and quantity of data in the source dataset and their geographical distribution. It means that ESI as well as other calculated parameters may be negatively affected by sampling bias. If samples from some habitats are underrepresented in the dataset, then some species with a broad niche may appear as more specialized [19]. Also, the results can be scale-specific because they depend on the size of the window in which the niche parameters are calculated. The applied ESI calculation methodology was designed to minimize sampling bias, and the same trend in the obtained results in both niche width and optimum shift on two spatial scales shows that the results are reliable (cf. Table 1, Figure 3, Figure 4 and Figure 5). In addition, the areas of Europe that are best covered by vegetation plots have higher ESI than others that are less covered (cf. Figure 2) [34]. Moreover, co-occurrence-based metrics are used to estimate potential shifts in species’ realized niche, taking into account many environmental gradients and incorporating not only the abiotic but also the biotic dimensions (species competition) [13,34].

In considering the differences between the margin and center of the range, the focus is on the western, northern and eastern edges but not the southern. The southern margin of the species’ range differs from the others for several reasons. The southern margin is sharper because it is geographically bound by the Mediterranean region. The Mediterranean climate is an obstacle to the spread of this species in the south. In addition, *A. clematitis* was present on the southern margin before it spread northward, probably as early as the Middle Ages [20,21]. Moreover, a limiting factor for the analysis of the southern margin is the low number and density of available plots from Central and Southern Italy, the south of the Balkan Peninsula and Turkey, which could also be due to the sporadic occurrence of the species. Considering these facts, it is very difficult to determine the southern margin of the species’ range. For this reason, it is necessary to study the ecology of *A. clematitis* at the southern edge with more field data.

### 4.5. Balkan Endemic Nephropathy

It is interesting that the geographical incidence of BEN largely corresponds to the geographical center of the distribution of *A. clematitis* [24,25,26]. Although our results show that the plot abundance of this species is not higher in the Balkans, an important finding is that, here, it occurs more often in man-made habitats, especially in the vegetation of arable lands and market gardens as well as anthropogenic herbaceous vegetation on the edge of agricultural fields (Figure 6 and Appendix A). Obviously, this ecological behavior allowed *A. clematitis* to come into contact with various crops, and, thus, the disease manifested itself more significantly here. Our results indicate that similar conditions appear in the north of Italy [54] and south of Ukraine (Figure 3 and Appendix A). However, contrary to Balkan countries, a higher incidence of chronic kidney disease and/or urothelial cancers was never reported from those regions. It would be interesting to investigate whether, in these regions, different harvesting and milling procedures disabled the commingling of *A. clematitis* seeds with wheat seeds. It is also possible that both in Italy and Ukraine, no one connected either chronic kidney disease or urothelial cancers with exposure to aristolochic acid. In such research, it should be taken into account that improved harvesting and milling technologies in Balkan countries have eliminated the contamination of wheat grain with *A. clematitis* seeds, thus decreasing exposure to aristolochic acid in home-baked bread and the subsequent development of BEN. In addition, nowadays, farmers purchase, for home consumption, flour and bread that is not contaminated with *A. clematitis* seeds. All these changes consequently resulted in a reduction in the prevalence of BEN. Even more, BEN completely disappeared in some former endemic villages [55].

## 5. Conclusions

Our results suggest that *A. clematitis* has a very broad ecological niche occurring in a range of different habitats and climatic conditions, with a trend of niche width decrease with the distance from the geographical center of the present distribution. The plant prefers more stable communities with less anthropogenic influence moving towards the margin of the distribution area. Specialization towards the marginal area is a result of the evolutionary history, which refers to recent anthropogenically induced spread from its original home range.

The high incidence of the species in the vegetation of arable lands and market gardens as well as anthropogenic herbaceous vegetation in the distribution center corresponds to the geographical incidence of BEN. The results of this research offer a significant contribution for understanding how the spatial distribution and ecological niche of *A. clematitis* were related to such a devastating disease as BEN. It also demonstrates how a multidisciplinary approach is needed for a complete understanding of environmental diseases.

## Figures and Tables

**Figure 1 plants-12-03022-f001:**
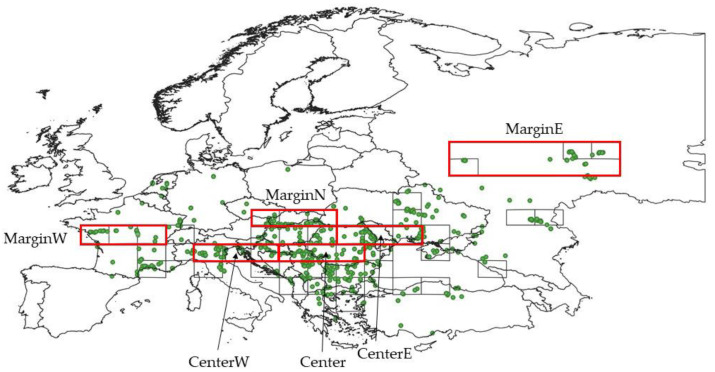
Localities of the plots after resampling (whole dataset, green dots), MGRS200 squares and regions used for calculating differences between central and marginal distribution range (bold rectangles).

**Figure 2 plants-12-03022-f002:**
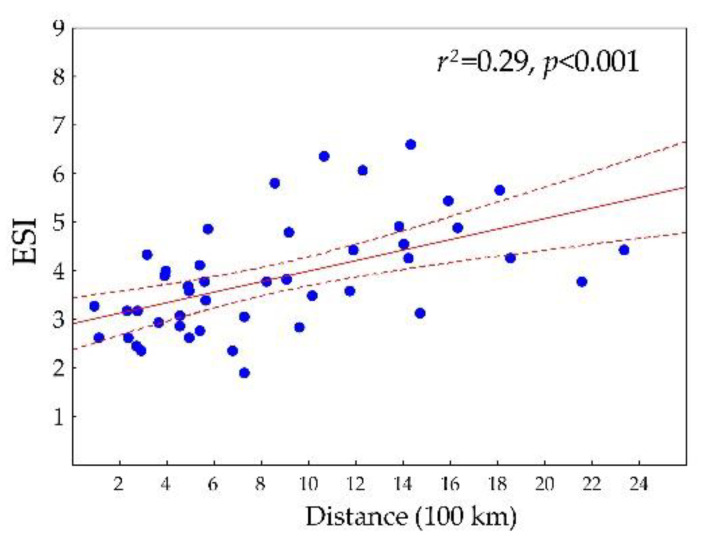
Relationships between distances of MGRS200 centroids from the geographic center and Ecological Specialization Indices (ESI).

**Figure 3 plants-12-03022-f003:**
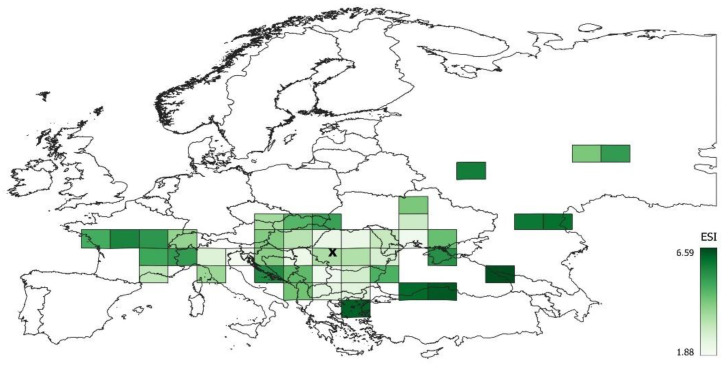
ESI values in studied MGRS200 cells. Letter x indicates geographical center of all analyzed vegetation plots.

**Figure 4 plants-12-03022-f004:**
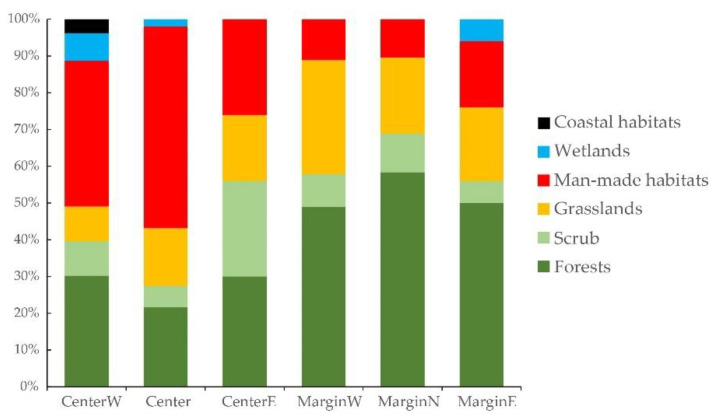
Percentage of particular habitat types (EUNIS level 1) in marginal and central regions.

**Figure 5 plants-12-03022-f005:**
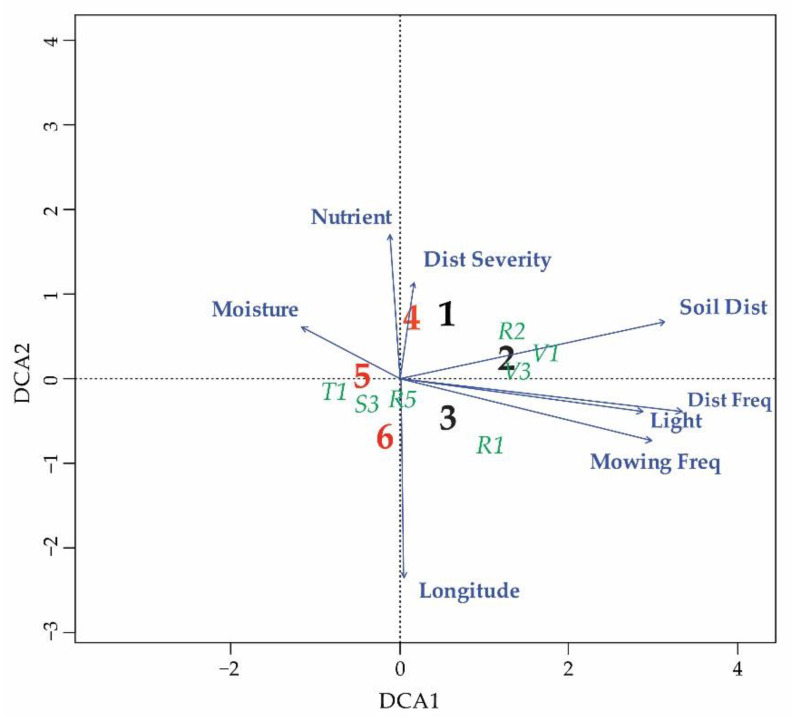
DCA ordination of the vegetation plots classified into three marginal and three central regions. Bolded red and black numbers correspond to centroids of all plots in particular region: 1—CenterW, 2—Center, 3—CenterE, 4—MarginW, 5—MarginN, 6—MarginE. Black numbers indicate central regions and red numbers indicate marginal regions. Mean Ellenberg indicator values and disturbance indicator values (bolded blue labels) as well EUNIS habitat types (green labels in italic) are passively projected. V1—arable land and market gardens, V3—artificial grasslands and herb-dominated habitats, T1—broadleaved deciduous forests, S3—temperate and Mediterranean-montane scrub, R1—dry grasslands, R2—seasonally wet and wet grasslands, R5—woodland fringes and clearings.

**Figure 6 plants-12-03022-f006:**
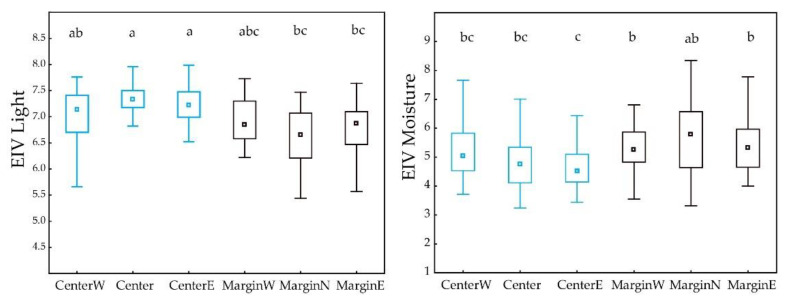
Differences in Ellenberg indicator values for light and moisture among marginal and central regions. Boxes show the 25–75% quartile range and the median value; whiskers indicate the range of values. Small letters indicate significant differences (*p* < 0.05).

**Figure 7 plants-12-03022-f007:**
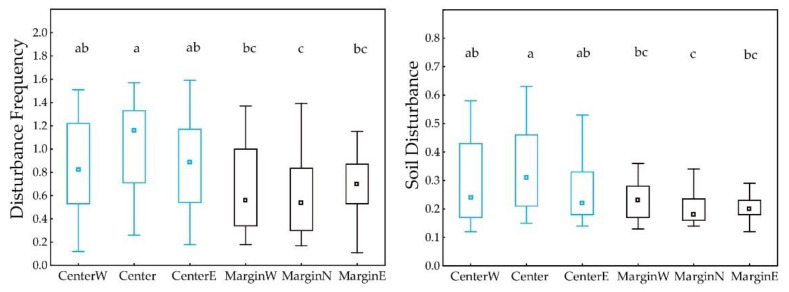
Differences in disturbance indicator values among marginal and central regions. Boxes show the 25–75% quartile range and the median value; whiskers indicate the range of values. Small letters indicate significant differences (*p* < 0.05).

**Table 1 plants-12-03022-t001:** Spearman correlations between ESI and medians and quartile ranges of variables describing ecological niche of *Aristolochia clematitis*. Only significant correlations are presented (*p* < 0.05).

Variable	Spearman R Between	
	ESI and Median	ESI and Quartile Range
Light	−0.35	-
Moisture	-	−0.48
Reaction	−0.33	-
Disturbance Severity	-	−0.49
Disturbance Frequency	−0.46	−0.50
Mowing Frequency	-	−0.45
Soil Disturbance	−0.38	−0.43
Perecentage of forest habitats	0.45	NA
Perecentage of man-made habitats	−0.41	NA

## Data Availability

Not applicable.

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
