# Peer review of "Different Ecological Niches of Poisonous Aristolochia clematitis in Central and Marginal Distribution Ranges—Another Contribution to a Better Understanding of Balkan Endemic Nephropathy"

_plants, 2023, doi:10.3390/plants12173022_

Round 1
Reviewer 1 Report
The paper titled "Different ecological niche of poisonous Aristolochia clematitis in central and marginal distribution range - another contribution to better understanding Balkan Endemic Nephropathy" presents an investigation into the ecological niche of Aristolochia clematitis in different parts of its distribution range and aims to establish a connection between the ecological niche and the prevalence of an endemic pathological condition.
The authors utilize data from the European Vegetation Archive to analyze and compute a synthetic index (ESI) that measures the ecological specialization of A. clematitis across different regions. They also examine the variation interval of several ecologically relevant parameters. The findings indicate a decrease in the ecological width as the distance from the geographical center increases.
The authors support their results for the global ecological niche with statistical tests, which should ideally be conducted for all measured or computed parameters (please refer to the annotations in the enclosed file).
While the investigation takes into account the marginal populations, it focuses only on the northern, eastern, and western margins, as well as the central region. Notably, there is no attempt to define a southern margin, and the central region is divided into three sub-regions (east, center, west). This choice of grouping may cause confusion, and the clarity of the paper would greatly benefit from organizing the plots into cells situated at the four cardinal points and the center. I strongly recommend re-grouping the plots.
The discussion section is generally appropriate and comprehensive, except for item 4.5. This particular chapter appears highly speculative and should be significantly reduced. Currently, there is an excessive emphasis on this aspect with limited support from the reported data. To delve deeper into this interesting topic, the authors are advised to establish a similar experimental design to collect and analyze data on the prevalence of Balkan Endemic Nephropathy (BEN) and the genetic makeup of the human populations. It would be appropriate to include this suggestion in the perspective for future research, without attempting to explain the differences between Italy, Croatia, and Ukraine. The authors should focus solely on their data and analysis.
In conclusion, the paper addresses an intriguing topic. However, the methodology requires revision for improved clarity as mentioned above, and the discussion section on BEN should be significantly reduced. The bibliography references are mostly adequate, but some are needed, as annotated in the enclosed file). The labels in some figures need improvement for better readability. While the English language usage is mostly grammatically correct, there is room for improvement in terms of clarity and fluency. Please refer to the annotated text for further comments.

While the English language usage is mostly grammatically correct, there is room for some improvement in terms of clarity and fluency. For instance, in the introduction the description of the Ukrainian settlement in Croatia is unclear; the lines 362-364 state that in marginal areas A. clematitis expands its niche to anthropogenic habitats, but earlier you stated that in marginal areas it prefers forests and not anthropogenic habitats. Other comments are annotated in the enclosed file.
Author Response
Response to Reviewer 1 Comments
We would like to thank the Reviewer for reading the manuscript carefully and giving very useful comments and suggestions. Response to Reviewer's general comments are given here. Detailed answers to the reviewer's comments given in the manuscript are given in the attached file.
Point 1: The paper titled "Different ecological niche of poisonous Aristolochia clematitis in central and marginal distribution range - another contribution to better understanding Balkan Endemic Nephropathy" presents an investigation into the ecological niche of Aristolochia clematitis in different parts of its distribution range and aims to establish a connection between the ecological niche and the prevalence of an endemic pathological condition.
The authors utilize data from the European Vegetation Archive to analyze and compute a synthetic index (ESI) that measures the ecological specialization of A. clematitis across different regions. They also examine the variation interval of several ecologically relevant parameters. The findings indicate a decrease in the ecological width as the distance from the geographical center increases.
The authors support their results for the global ecological niche with statistical tests, which should ideally be conducted for all measured or computed parameters (please refer to the annotations in the enclosed file).
Response 1: We accepted the Reviewer's suggestion and performed additional statistical tests. A detailed explanation is given in the attached file.
Point 2: While the investigation takes into account the marginal populations, it focuses only on the northern, eastern, and western margins, as well as the central region. Notably, there is no attempt to define a southern margin, and the central region is divided into three sub-regions (east, center, west). This choice of grouping may cause confusion, and the clarity of the paper would greatly benefit from organizing the plots into cells situated at the four cardinal points and the center. I strongly recommend re-grouping the plots.
Response 2: We completely understand the reviewer's comment. Although the differences between the center and the margin of distribution area are already clear from the analysis of all cells, due to the large number of these cells, we wanted to present the results more clearly and transparently by grouping some of them, or other words by directly comparing the margin and the center of the distribution area. We thought a lot about the selection of groups and it is always arbitrary, but regardless of how we did it, the results do not differ significantly.
We partially agree that we could have chosen one central region instead of three, however, it seems to us that in this way a better understanding of the results is possible, and most importantly, they do not differ if we took only one group.
Regarding the southern margin, your comment pointed out that we did not explain it well in the manuscript, which we have now tried to improve. The southern margin of the species range is different from the others for several reasons. Namely, it is sharper because it is limited geographically by the Mediterranean Sea but also by an unfavorable Mediterranean climate, which represent an obstacle for the spread of the species to the south. In addition, according to the available literature, the species was present in that area even before spreading to the north (probably during the Middle Ages). However, a limiting factor for the analysis of the southern edge is the low number and density of available plots from central and southern Italy, the south of the Balkan Peninsula and Turkey. For this reason, we are of the opinion that the evaluation of the southern margin should be left for future research. We tried to explain all of the above in detail in the discussion chapter (L442-453).
Point 3: The discussion section is generally appropriate and comprehensive, except for item 4.5. This particular chapter appears highly speculative and should be significantly reduced. Currently, there is an excessive emphasis on this aspect with limited support from the reported data. To delve deeper into this interesting topic, the authors are advised to establish a similar experimental design to collect and analyze data on the prevalence of Balkan Endemic Nephropathy (BEN) and the genetic makeup of the human populations. It would be appropriate to include this suggestion in the perspective for future research, without attempting to explain the differences between Italy, Croatia, and Ukraine. The authors should focus solely on their data and analysis. In conclusion, the paper addresses an intriguing topic. However, the methodology requires revision for improved clarity as mentioned above, and the discussion section on BEN should be significantly reduced.
Response 3: We accepted the Reviewer's suggestion and significantly reduced this chapter. We tried to write it less speculatively, and have provided perspective for future research.
Point 4: The bibliography references are mostly adequate, but some are needed, as annotated in the enclosed file).
Response 4: We checked and corrected all mistakes in the bibliography. A detailed explanation is given in the attached file.
Point 5: The labels in some figures need improvement for better readability.
Response 5: We accepted all the suggestions and corrected the figures and their captions. A detailed explanation is given in the attached file.
Point 6: While the English language usage is mostly grammatically correct, there is room for improvement in terms of clarity and fluency.
Response 6: The English language has been reviewed and corrected by a native speaker
Point 7: Please refer to the annotated text for further comments.
Response 7: We accepted all other Reviewer's comments, A detailed explanation are given in the attached file.

Reviewer 2 Report
This is interesting research and very meaningful work. I would like to
make some small suggestions.
1. In the Materials and Methods section, add the research framework and
formulate a technical roadmap so that readers can clearly understand the
ideas and framework of this research.
2. After the Discussion section, a Conclusion section should be added to
make the structure of the manuscript more complete.
Moderate editing is required for the English language.
Author Response
Response to Reviewer 2 Comments
Point 1: In the Materials and Methods section, add the research framework and formulate a technical roadmap so that readers can clearly understand the ideas and framework of this research.
Response 1: Research framework and study roadmap were added in a paragraph at the beginning of Materials and methods section (L138-145).
Point 2: After the Discussion section, a Conclusion section should be added to make the structure of the manuscript more complete.
Response 2: Conclusion section was added
Reviewer 3 Report
I have found Your article as very valuable work. It has a great novelty potential, and it deals with the new findings about association of poisoneous Aristolochia clematitis plant with Balkan Endemic Neprophaty problem.
However, I suggest to shorten the Introduction, since in my opinion it is now a little too long. Also plesae change the reference number from 20 to 19 on page 2 (line 89), since the Authors Zeleny and Chitry were assigned with number 19 in the References.
Author Response
Response to Reviewer 3 Comments
Point 1: I suggest to shorten the Introduction, since in my opinion it is now a little too long.
Response 1: The introductory section has been shortened by 110 words (9 lines).
Point 2: Also plesae change the reference number from 20 to 19 on page 2 (line 89), since the Authors Zeleny and Chitry were assigned with number 19 in the References.
Response 2: Corrected
Round 2
Reviewer 1 Report
I appreciate the amendments and the paper can be published in its current form.